# Text Rendering Strategies for Pixel Language Models

**Jonas F. Lotz**[†,‡]   **Elizabeth Salesky**[⋆]   **Phillip Rust**[†]   **Desmond Elliott**[†]

[†]Department of Computer Science, University of Copenhagen
[‡]ROCKWOOL Foundation Research Unit
[⋆]Johns Hopkins University
jonasf.lotz@di.ku.dk

## Abstract

Pixel-based language models process text rendered as images, which allows them to handle any script, making them a promising approach to open vocabulary language modelling. However, recent approaches use text renderers that produce a large set of almost-equivalent input patches, which may prove sub-optimal for downstream tasks, due to redundancy in the input representations. In this paper, we investigate four approaches to rendering text in the PIXEL model (Rust et al., 2023), and find that simple character bigram rendering brings improved performance on sentence-level tasks without compromising performance on token-level or multilingual tasks. This new rendering strategy also makes it possible to train a more compact model with only 22M parameters that performs on par with the original 86M parameter model. Our analyses show that character bigram rendering leads to a consistently better model but with an anisotropic patch embedding space, driven by a patch frequency bias, highlighting the connections between image patch- and tokenization-based language models.

## 1 Introduction

There is a growing movement in NLP towards tokenization-free methods (Clark et al., 2022; Xue et al., 2022; Yu et al., 2023) including pixel-based representations of text (Salesky et al., 2021, 2023; Rust et al., 2023; Tschannen et al., 2023). It has been shown that these tokenization-free methods can readily handle unseen languages and that they are more robust to noise attacks than tokenization-based models. In addition, pixel-based approaches can effectively exploit visual similarities between characters and scripts because they allow for complete parameter sharing across all inputs, making them a promising direction for multilingual NLP.

Previous work on pixel-based models segments the rendered text into either consecutive patches (Rust et al., 2023; Tschannen et al., 2023) or with

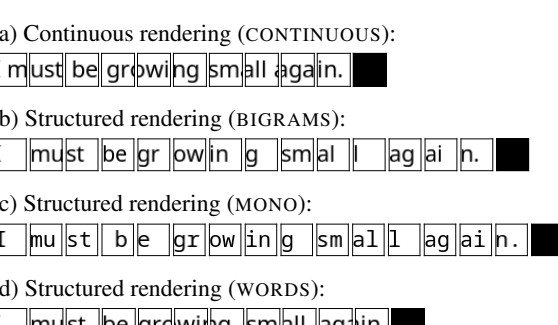

Figure 1: Examples of rendering strategies for the sentence "*I must be growing small again.*" from Carroll (1865). Black patches mark the end of a sequence, following Rust et al. (2023).

a sliding window (Salesky et al., 2021, 2023) as in speech processing. Although the proposed approaches have the appealing properties of yielding compact and transferable representations, they also result in a very large input space because there is no unique way to represent lexical units. As a consequence, pixel-based models could observe a new set of *image* representations with every new sentence, which adds redundancy in the input space and is sub-optimal for developing contextual *language* representations. We refer to these unstructured rendering strategies as CONTINUOUS and illustrate the point qualitatively in Figure 1 and Figure 2, and quantitatively in Figure 3. In this work, we ask whether structuring the input, which leads to more frequent parameter updates through now-unique word representations, would enable pixel-based models to develop a deeper understanding of context and semantics. We then propose rendering strategies structured around providing the model with a compressed input space.

We demonstrate how enforcing a BIGRAMS-structured rendering strategy leads to both a more capable and data-efficient model: when evaluated on semantic sentence-level tasks, we find that a 22M parameters model performs

(a) Most frequent patches with CONTINUOUS rendering:

the the the the the the he the he he

(b) Most frequent patches with BIGRAMS rendering:

e th in d s an of on re ,

Figure 2: A continuous rendering strategy results in many uniquely-valued image patches for similar inputs, while structured rendering (here, BIGRAMS) regularises and compresses the potential input space.

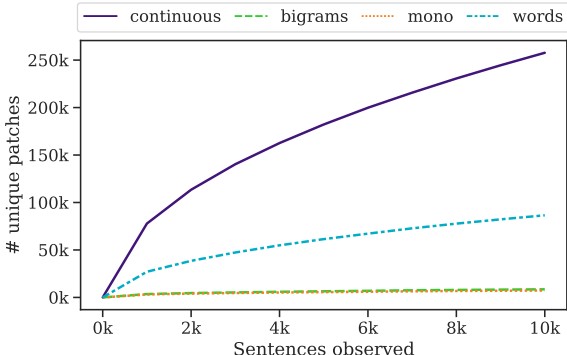

Figure 3: Number of unique image patches observed as a function of training data sequences. Structured rendering results in greater representational efficiency.

competitively with the unstructured original at 86M parameters, and that scaling back up to 86M parameters narrows the performance gap to BERT (Devlin et al., 2019) trained on the same data. In subsequent analyses, we find that the added input structure provokes a clear visual token frequency bias in the learned embedding space. While also found in BERT, frequency biases have been shown to degrade the quality of embedding spaces when word representations are not only determined by semantic relations but also by the number of model updates (Gong et al., 2018; Gao et al., 2019; Fuster Baggetto and Fresno, 2022). We show that frequent words have more context-specific representations than infrequent words, especially in the upper layers. Finally, we show that PIXEL models acquire a non-trivial semantic understanding during pretraining, but that their sentence representations are easily influenced by this frequency bias. We release all models[1] and code[2] for pretraining and finetuning.

## 2 Background: modelling text as images

We build upon the general-purpose language encoder framework presented in Rust et al. (2023): PIXEL is a text autoencoder which builds on the Masked Autoencoding Vision Transformer (ViT-MAE; He et al., 2021) and is similarly pretrained with a masked reconstruction objective. However, instead of patches from natural images of objects (Deng et al., 2009), the patches now contain images of text. To go from text to images of text, PIXEL relies on a rendering library (PangoCairo)[3] to produce a sequence-level image which is sliced into image patches of size $16 \times 16$ pixels. The sequence-length maximum of 529 patches approximately equals the memory requirements of BERT,

the closest benchmark for PIXEL. By using the Google Noto font family which supports the majority of Unicode codepoints,[4] the renderer supports all languages that can currently be typeset.

Before the first layer of the PIXEL model, image patches are linearly projected to obtain a sequence of patch 'embeddings'. During pretraining, 25% of embeddings are masked in spans of up to 6 patches and only the unmasked patches with a prepended CLS embedding are passed through the encoder. After replacing the masked embeddings amidst the encoder outputs, relying on fixed sinusoidal position embeddings for ordering information, the decoder predicts the pixel values of solely the masked patches. To later finetune the encoder on a classification task, the decoder can be replaced with a task-specific head and the masking ratio set to 0%.

## 3 Structured rendering

Previously proposed approaches to rendering text as images render full sequences of text and segment into either consecutive patches (Rust et al., 2023; Tschannen et al., 2023) or with a sliding window (Salesky et al., 2021, 2023). These CONTINUOUS strategies result in a significant number of uniquely-valued patches, many of which may be observed only once during training. We depict this redundancy in Figure 2 and quantify it in Figure 3, showing how similar text inputs result in unique visual representations.

We compare four rendering strategies: the original unstructured (CONTINUOUS), and three structured (WORDS, MONO, BIGRAMS), as depicted in Figure 1. To render WORDS we separate seg-

---

[1] https://huggingface.co/Team-PIXEL
[2] https://github.com/xplip/pixel/tree/TextRenderingStrategies
[3] https://docs.gtk.org/PangoCairo
[4] https://fonts.google.com/noto

ments with additional whitespace[5] such that new segments begin at the beginning of the next image patch, regulating possible spatial variation. BI-GRAMS, rendering two characters per image patch, is chosen to be widely applicable, without knowledge of word or morphemic segmentation (Mielke et al., 2021; Keren et al., 2022). More specifically—consider the word pairs ⟨"grow", "growing"⟩ and ⟨"growing", "walking"⟩—the BIGRAMS renderer will produce an overlap of image patches (underlined) for both pairs while the same extent is not guaranteed with WORDS-level rendering as it is regulated by character width. The choice of character $(n = 2)$-grams is motivated by what generally fits within a $16 \times 16$ pixels image patch in the setup from Rust et al. (2023). MONO instead applies monospaced fonts where each character is a fixed width; depending on font size, this may result in character bigram patches without breaks within characters, but this is not guaranteed. The main difference between BIGRAMS and MONO is that MONO simply slides across the sentence, two characters at the time, yielding two ways to represent a word whereas BIGRAMS renders the words and then pads with whitespace, ensuring unique inputs.[6]

As seen in Figure 3, the structured rendering strategies result in a greatly compressed input space as measured by the number of unique image patches processed by the model, but Figure 1 reveals that it comes at the cost of longer sequence lengths. While the rendering strategies we propose were not specifically designed for English, they may not equally generalise to other languages or scripts. We further discuss the representational efficiencies of these strategies in § A.1 and limitations to generalisability under Limitations.

## 4  Model scale variants

Recall from Figure 3 that CONTINUOUS rendering produces a significantly larger set of unique image patches compared to other approaches. A consequence of this is that models must learn to encode many almost-identical visual representations, which may be wasteful, both in terms of parameters and training efficiency. Therefore, we hypothesise that PIXEL models that operate over fewer unique image patches can be scaled down without sacrific-

| Model | Enc$_L$-Dec$_L$ | Hid | MLP | Att | $|\theta|$ |
|-------|-----------------|-----|-----|-----|------------|
| BASE  | 12-8 | 768 | 3072 | 12 | 86M |
| SMALL | 12-4 | 384 | 1536 | 6 | 22M |
| TINY  | 12-2 | 192 | 768 | 3 | 5.5M |

Table 1: Details of PIXEL model scale variants.

ing performance. While "Base" models and larger ones are widely used for their strong performance, proven scaling laws (Touvron et al., 2021; Zhai et al., 2021) enable greater experimentation and model development at smaller scale (Ivgi et al., 2022), which is both more environmentally friendly (Strubell et al., 2019; Bender et al., 2021; Hershcovich et al., 2022) and facilitates contributions with limited computational resources.

With this in mind, we propose two smaller architectures which we will compare across downstream tasks in § 5. Our BASE model architecture is directly adopted from ViT (Dosovitskiy et al., 2021) and PIXEL, and we add two more compact SMALL and TINY model variants, as described in Table 1. The configurations of the smaller models are based on the ViT variants presented in Zhai et al. (2021). Following the scaling experiments in He et al. (2021), indicating that shallow decoders of as small as 2 layers can be sufficient for ViT-MAEs, we apply a scheme of halving the number of decoder layers at every scale reduction.

## 5  Experiments

We pretrain SMALL models with the proposed rendering strategies. The models are then evaluated on dependency parsing (UDP) with data from Universal Dependencies v2.10 treebanks (Zeman et al., 2022; Nivre et al., 2020) and GLUE (Wang et al., 2018), exploring the models' capabilities at syntactic processing on the word level and semantic processing on the sentence level.

### 5.1  Pretraining

We pretrain all models on the English Wikipedia and Bookcorpus (Zhu et al., 2015) data used by Rust et al. (2023) for direct comparison with PIXEL and BERT, which results in ~16.8M training examples. We follow the suggested hyperparameters used for PIXEL with the exception of batch size. The smaller architectures of SMALL and TINY allow for larger batch sizes, which we double from 256 examples to 512 and 1024, respectively. We then halve the number of pretraining steps accord-

---

[5]We render whitespace at minimum 3 pixels wide, sometimes resulting in a blank patch between tokens in structured inputs.

[6]As an example, "be" in Figure 1 is split into 2 image patches with MONO rendering. Depending on the context, it could also be represented in a single image patch.

| | Structure | | | | Scale | | | | | |
| | UDP | GLUE | | | UDP | | GLUE | | TyDiQA-GoldP | |
| *Renderer* | *Avg.* | *Avg.* | *Variant* | $|\theta|$ | *Avg.* | $\Delta\mu$ | *Avg.* | $\Delta\mu$ | *Avg.* | $\Delta\mu$ |
|---|---|---|---|---|---|---|---|---|---|---|
| CONTINUOUS | 76.2 | 71.0 | TINY | 5.5M | 72.0 | −0.3 | 66.5 | +12.7 | 41.6 | +4.9 |
| BIGRAMS | 76.1 | 75.4 | SMALL | 22M | 76.1 | −0.1 | 75.4 | +4.4 | 50.8 | +2.0 |
| MONO | 75.9 | 74.4 | BASE | 86M | 75.5 | −0.6 | 78.0 | +3.9 | 52.8 | +0.5 |
| WORDS | 76.6 | 74.7 | BERT | 110M | 50.5 | — | 80.0 | — | 51.5 | — |

Table 2: **Structure** (left): averaged results for SMALL-models comparing downstream performance on UDP and GLUE following the different rendering strategies. **Scale** (right): averaged results across model scales using the BIGRAMS rendering structure. $\Delta\mu$ is the difference in average performance between BIGRAMS and CONTINUOUS rendering for a given model scale. BERT results are marked in grey to visually distinguish from pixel-based models.

ingly from 1M to 500k and 250k in order to train for the same number of epochs as PIXEL (~16 epochs, but varying slightly due to differing sequence lengths per rendering strategy).

Pretraining BASE takes 8 days on $8 \times 40$GB Nvidia A100 GPUs, while in comparison, pretraining SMALL takes less than 48 hours on $8 \times 40$GB Nvidia A100 GPUs, and TINY less than 24 hours. Loss trajectories for the different rendering strategies are in line with their representational efficiency (Figure 3), indicating that structured rendering may make the masked reconstruction task more data-efficient, achieving a low loss in fewer steps (see § A.2: Figure 10).

## 5.2 Finetuning

To finetune our models for classification tasks we replace the decoder used for pretraining with a task-specific classification head. We do not search for more optimal hyperparameters than those used for PIXEL with the exception of the learning rate; we find that the more compact architectures often benefit from a slightly higher learning rate.[7]

We follow the same protocol during finetuning as done for PIXEL: for word-level tasks we obtain the rendered image patch indices for every word and as a consequence, the CONTINUOUS strategy becomes identical to the WORDS structure when finetuning on UDP. § 6.1 further investigates the consequence of a mismatch between how the data is structured during pretraining and finetuning. When finetuning on GLUE the structure follows what was seen during pretraining for all rendering strategies. Reported performances for BERT and PIXEL are taken from Rust et al. (2023).

---

[7] We search the space $\{1e-5, 3e-5, 5e-5, 7e-5, 9e-5\}$ and report the average over 3 seeds.

## 5.3 Rendering strategies

We present averaged results comparing the rendering strategies in the left part of Table 2. Detailed results for each downstream task are presented in Table 4 and Table 5 in the appendix. For UDP we find that the WORDS structure slightly outperforms BIGRAMS and MONO on this word-level task. When comparing the WORDS and CONTINUOUS strategies we get a first hint as to the importance of including structure during pretraining as well, keeping in mind that the rendering structure is the same for both strategies when finetuning on UDP. For GLUE we see a large increase in performance when rendering with any structure and especially BIGRAMS. We attribute the difference in performance between BIGRAMS and MONO to the unique word representations with BIGRAMS, as discussed in § 3.

We find that BIGRAMS is the best performing structure on average, even slightly outperforming the 86M parameters PIXEL (average UDP: 76.1; average GLUE: 74.1) with only ¼ its model parameters. We provide an investigation into the mechanisms that enable this improved performance on GLUE in § 6.4. Next we pretrain TINY and BASE model variants with BIGRAMS rendering to evaluate performance at different model scales.

## 5.4 Model scaling

The right part of Table 2 compares the different model scales all following a BIGRAMS rendering strategy. Detailed results are likewise presented in Table 4, Table 5, and Table 6 in the appendix. We find that the TINY configuration performs competitively on the word-level tasks considering its only 5.5M parameters, but has a larger gap up to SMALL and BASE on the sentence-level GLUE tasks. SMALL proves to be a good trade-off between scale and performance where it is not far behind BASE on GLUE and even slightly

outperforms on UDP.[8] BASE comes a step closer to closing the gap in performance up to BERT on GLUE. Comparing to the performance following a CONTINUOUS rendering strategy, summarised as the difference in average performance ($\Delta\mu$), it is clear that the more compact the model size, the greater the benefit from structured rendering.

To verify that BIGRAMS rendering does not degrade the performance on *multilingual* sentence-level tasks across different scripts and morphologies, we also include results on TyDiQA-GoldP (Clark et al., 2020).[9] Again we find that SMALL performs competitively considering its size.

## 6 Ablations and supplementary analyses

In this section we investigate how BIGRAMS rendering changes the model compared to CONTINUOUS. For clarity in what follows, we refer to the BASE model with BIGRAMS rendering from § 5.4 as BASE-BIGRAMS and keep referring to the original model from Rust et al. (2023) as PIXEL.

### 6.1 When does rendering structure matter?

Having established that a structured rendering strategy leads to improved downstream performance, we further investigate *when* it is needed: is it sufficient to finetune with structure or does the model develop strategy-specific features during pretraining? We analyze this by comparing rendering strategies between pretraining and finetuning.

The results in Table 3 for GLUE show that a mismatch leads to lower downstream performance for both strategies, with BIGRAMS → CONTINUOUS being the most harmful, perhaps unsurprisingly. This result does not align with the finding for UDP in § 5.3 where CONTINUOUS overcomes the change to WORDS-structured rendering. It may indicate that the lower-level UDP tasks are easier for PIXEL-based models than the high-level GLUE tasks (Lauscher et al., 2020). This is in line with the relatively good performance for TINY-BIGRAMS on UDP.

To emphasize the increase in performance on semantic tasks with BIGRAMS rendering, we

| RENDERER | | GLUE |
|---|---|---|
| Pretraining | Finetuning | *Avg.* |
| BIGRAMS | BIGRAMS | 75.4 |
| CONTINUOUS | CONTINUOUS | 71.0 |
| CONTINUOUS | BIGRAMS | 61.1 |
| BIGRAMS | CONTINUOUS | 53.0 |

Table 3: Rendering strategy combinations between pretraining and finetuning with SMALL models. For GLUE, matching pretraining structure is most effective.

demonstrate that BASE-BIGRAMS outperforms PIXEL by 3.6 points on average on MasakhaNER (Adelani et al., 2021), a named entity recognition benchmark for 10 African languages. This further illustrates the potential of PIXEL-based models for modelling low-resource languages. Detailed results are presented in Table 7 in the appendix. We next turn our attention to *how* BIGRAMS rendering enables better performance on semantic tasks.

### 6.2 Contextual representations

The extent to which language models capture semantic information is partly determined by their ability to contextualise text (Peters et al., 2018). We therefore analyse how capable BASE-BIGRAMS is at producing contextualised word representations. We use the Words in Context dataset (WiC; Pilehvar and Camacho-Collados, 2019) of sentences that contain target words (noun or verb) in either a similar (True) or different (False) context across sentence pairs.[10] We compute the mean hidden state output over all tokens associated with the target word to obtain a representation. We infer that there is contextualisation if the model generates representations of a target word from different contexts with a low cosine similarity compared to target words in similar contexts. We report this indication of contextuality for each layer of the model, including the input layer, to better understand the properties of the different layers. Similarities between randomly chosen words from random examples (Random) are included as a baseline.[11]

Figure 4a plots the resulting distributions of similarities. We see that representations of target words from similar contexts have a higher cosine similarity than from different contexts, though with

---

[8]We expect that BASE could prevail and would benefit from a wider search for optimal hyperparameters during finetuning.

[9]With the CONTINUOUS rendering strategy, answer spans are extracted such that the answer may include leading or trailing characters when there is no exact mapping from a word to an image patch index. Therefore, we did not include TyDiQA-GoldP in the comparison in § 5.3. More details can be found in Rust et al. (2023). We discuss limitations to answer span extraction with BIGRAMS rendering in § A.4.

[10]Target words are not necessarily identical across sentence pairs and can vary e.g. in conjugation or number.

[11]It is not possible to obtain an exact mapping from words to neat image patch indices following the CONTINUOUS rendering strategy so we do not present this analysis for PIXEL.

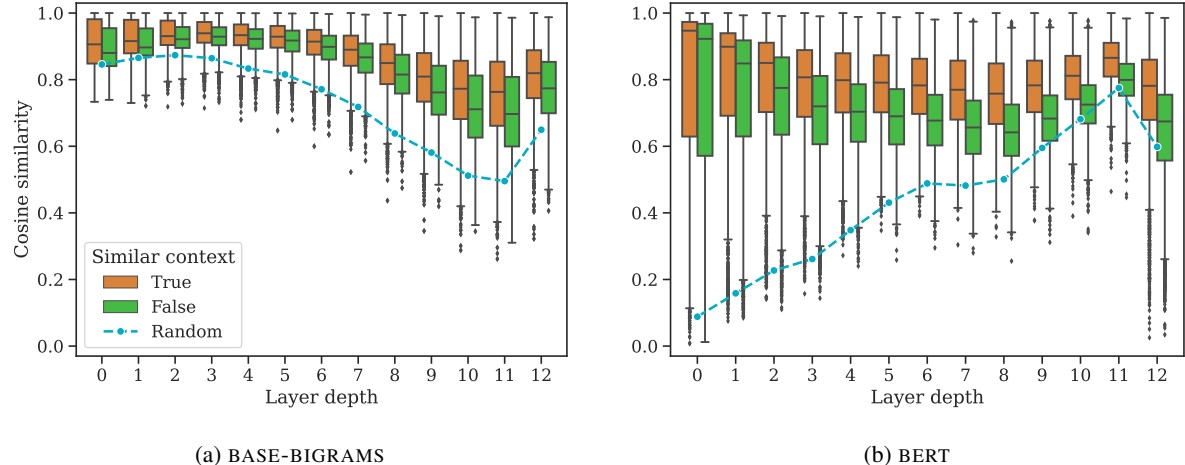

(a) BASE-BIGRAMS

(b) BERT

Figure 4: Distributions of cosine similarities for verbs and nouns from the WiC dataset across model layers 0-12, layer 0 being the input layer. Every example presents a target word in either a similar or different context across a sentence pair. The representation of the target word is computed as the mean hidden state output over the corresponding tokens. We generally see that BASE-BIGRAMS encodes target words in a similar context as more similar. The median cosine similarity between random words from random sentences are shown as a baseline.

a considerable overlap, and higher for different contexts than for random. When comparing to BERT in Figure 4b, there is a clear difference in the similarity compared to random words. The difference in similarity between similar and random words gradually increases throughout the BASE-BIGRAMS model, until the final layers, whereas the difference steadily decreases throughout the model for BERT. Given the shared image patch embedding layer in PIXEL-based models, random words are more similar to each other at the input layer when modelled as images than entries in a vocabulary.

Taken together, these plots suggest that a PIXEL-based language model is capable of forming contextualised word representations and that these are more context-specific in upper layers, though not as fine-grained as seen for BERT.

## 6.3 Token frequency and similarity

The degree of cosine similarity between random words observed in Figure 4a encourages us to assess the isotropic nature of the model (Ethayarajh, 2019; Rajaee and Pilehvar, 2021). The high cosine similarities suggest that the word representations are not evenly distributed with respect to direction in the embedding space, but instead appear to be anisotropic. When learned vector representations populate a narrow cone in the embedding space, this geometric alignment leads to an overestimation of their similarity (Gao et al., 2019), which is not an expected property of

an expressive word embedding space (Arora et al., 2016; Mu and Viswanath, 2018).[12]

Recent work has shown that Transformer-based language models can develop a representation bias driven by token frequency, where low-frequency tokens are clustered together in the embedding space, leading to anisotropy in the model (Gao et al., 2019; Fuster Baggetto and Fresno, 2022; Jiang et al., 2022). This bias leads to poor word contextualisation because the learned vector positions of low frequency words have not moved far from their random initialisation. Thus, their embeddings are not sufficiently distinct from unrelated words with similarly low token frequency (Gong et al., 2018; Cai et al., 2021). Tokens with a higher frequency, and thus more parameter updates, can move further in the embedding space from their initialisation and become more *semantically meaningful*. Consequently, we hypothesise that compressing the input space in the form of structured rendering allows the model to build more contextualised word representations through more frequent parameter updates.

We investigate this by sampling inputs that were seen during pretraining with high and low frequency. Specifically, we take the 100 most fre-

---

[12]Following Cai et al. (2021) this *global* estimate of ansiotropy does not rule out the possibility of distinct and locally isotropic clusters in the embedding space. Ding et al. (2022) show that isotropy calibration methods (Gao et al., 2019; Wang et al., 2020; Li et al., 2020) do not lead to consistent improvements on downstream tasks when models already benefit from local isotropy. We leave this direction for PIXEL to future research.

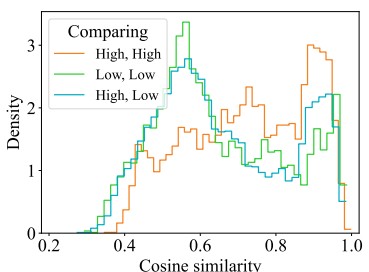 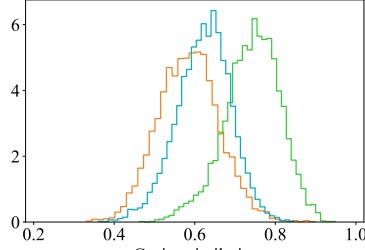 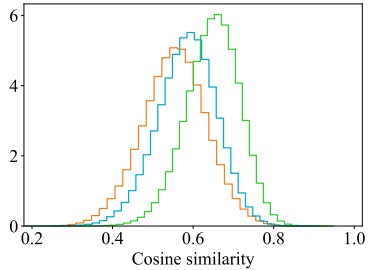

(a) Words in isolation, PIXEL   (b) Words in isolation, BASE-BIGRAMS   (c) Words in context, BASE-BIGRAMS

Figure 5: Distributions of cosine similarities within samples of high-frequency words (High), low-frequency words (Low), or between the two samples. Rendering with BIGRAMS structure leads to less directionally aligned vector representations of frequent words that have seen more updates during pretraining compared to infrequent words.

quently occurring words from the Wikipedia corpus that was seen during pretraining and 100 words that occur around 1000 times (rank ≈ 50k).[13] We first render each word from the two frequency samples in isolation. We then include a comparison to words in context across 100 unique sentences per word with BASE-BIGRAMS.[14]

We plot the distributions of cosine similarities between representations from the last encoder layer, where we expect embeddings from both models to be contextualised. Comparing the plots from the two rendering strategies, summarised in Figure 5, the effect of pretraining with a smaller set of unique tokens becomes clear: for PIXEL the distribution appears as mixtures with a larger distribution mass at higher values of cosine similarity from comparing high-frequency words to other high-frequency (excluding self-similarity for now) than when comparing low-frequency to other low-frequency. For BASE-BIGRAMS the frequent words both in isolation and in-context are less directionally aligned with each other compared to the infrequent, which is in line with the *representation degeneration problem* from Gao et al. (2019) and more frequent updates leading to better contextualisation. Figure 6 visualises the in-context representations in 2 dimensions using t-SNE (van der Maaten and Hinton, 2008) and provides an additional indication of more frequent words having less locally compact representations.[15]

We expect that in-context representations from PIXEL also qualitatively resembles Figure 5a but cannot easily demonstrate this due to the

aforementioned challenges in aligning patch embeddings with CONTINUOUS rendering.

## 6.4 Frequency bias and semantic modelling

While there is less evidence of representation degeneration with CONTINUOUS rendering, it is likely that the poorer performance on GLUE in § 5.4 is caused by PIXEL seeing too many different patches too few times. This is a direct consequence of the multitude of ways that similar inputs can be rendered by the CONTINUOUS approach. However, the drop in performance when mismatching the rendering strategies in § 6.1 for CONTINUOUS → BIGRAMS demonstrates that the model has developed a set of strategy-specific expectations and features that are not easily updated. In fact, the new rendering strategy for finetuning introduces a set of patches that likely never escape the low-frequency domain and therefore remain poorly contextualised. Signs of a token frequency bias has also been found in BERT (Fuster Baggetto and Fresno, 2022).

We lastly assess the connection between visual token frequency and downstream semantic performance. With BERT, high-frequency words have the most context-specific representations (Ethayarajh, 2019), and upper-layer representations of low-frequency words are influenced more by their context than frequent words (Voita et al., 2019). Following Ethayarajh (2019), we see that this applies to BASE-BIGRAMS as well (illustrated in Figure 7 and discussed in greater detail in § A.5). We expect that sentences that only vary in being cased or uncased would result in different representations when lowercase appears more frequently (for most words). This demonstrates the impact of observed token frequency on semantic modelling and is in line with observed biases in BERT's embedding space (Jiang et al., 2022).

---

[13]Excluding punctuation and numbers.

[14]Recall from § 6.2 that the CONTINUOUS rendering strategy by design makes an exact mapping from words in a sentence to neat image patch indices unattainable.

[15]Plotting the first 2 singular values from a singular value decomposition gives the same qualitative indications.

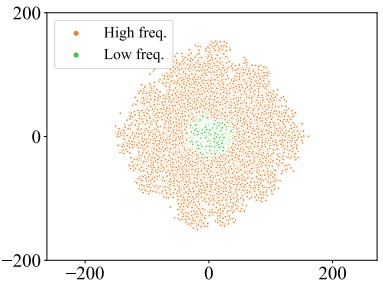
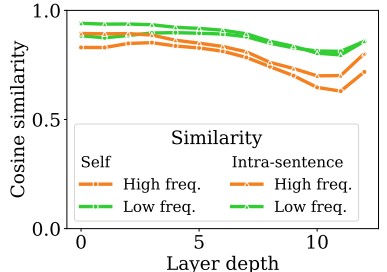
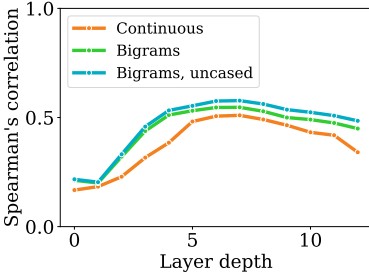

Figure 6: t-SNE plot of the output embeddings of high- and low-frequency words in context from BASE-BIGRAMS. Low-frequency words cluster tightly in this space.

Figure 7: Self- and intra-sentence similarity from BASE-BIGRAMS. High-frequency words are the most context-specific; low-frequency words are influenced by their context.

Figure 8: Evaluation performance on STS-B. Uncased sentences yield better performance than the original with BASE-BIGRAMS; the effect is less clear for PIXEL (not shown).

We rely on the Semantic Textual Similarity Benchmark (STS-B; Cer et al., 2017) also found in GLUE for this assessment. We measure the cosine similarity between sentence representations[16] and plot its correlation with the gold standard similarity scores as the measure of performance. Figure 8 proves that both CONTINUOUS and BIGRAMS rendering during pretraining lead to non-trivial semantic modelling capabilties. At peak performance, around the middle layers, the increase from simply ensuring that all words are uncased is roughly the same as the increase from PIXEL to BASE-BIGRAMS. This resembles how frequent and infrequent tokens have unequal influence *on* their context in BERT (Voita et al., 2019).

Seeing that BASE-BIGRAMS exhibits similar representational traits to that of BERT, future work could aim for more semantically capable PIXEL-based models by generalising advances found for tokenizer-based models (Gao et al., 2021).

# 7   Related work

Recent work on pixel-based language modelling has demonstrated how visual language understanding can be achieved through pixels only (Lee et al., 2022), observed that the visual similarity of languages plays an important role in cross-lingual transfer (Rahman et al., 2023), and shown how unifying the modalities for text and images allow a single encoder to perform multimodal tasks (Tschannen et al., 2023). By relying on bytes directly, the unification of modalities can be taken even further (Jaegle et al., 2021; Horton et al., 2023; Yu et al., 2023). The work most closely

related to ours, after Rust et al. (2023), is the work on machine translation with pixel representations (Salesky et al., 2021, 2023). A detailed discussion of previous pixel-based approaches can be found in Rust et al. (2023, § 5). Where PIXEL laid the foundation for general-purpose language encoding with pixel-based representations, this work takes the first step towards hypothesis-driven improvements without adding additional data (Yang et al., 2019) or scaling up the model (Conneau and Lample, 2019). Though it is possible that competitive performance could be achieved by a model with CONTINUOUS rendering by pretraining on more data for more steps (Liu et al., 2019).

Our addition of BIGRAMS structure resembles the addition of optional but hugely beneficial ($n = 4$)-grams in the character-based CANINE model (Clark et al., 2022). While character-level $n$-gram models (Wieting et al., 2016; Bojanowski et al., 2017) have been succeeded by Transformer-based language models, character-level features remain valuable as they are less sparse and more robust to misspellings than word $n$-grams, and remain useful for especially morphologically rich languages (Garrette and Baldridge, 2013; Kulmizev et al., 2017). Previous work have hypothesised that character-level models would be more suitable than subword-based for modelling morphologically-rich languages (Tsarfaty et al., 2020; Keren et al., 2022), but a semantically capable design has proven non-obvious (Ma et al., 2020; Keren et al., 2022; Nzeyimana and Niyongabo Rubungo, 2022; Sun et al., 2023). We see potential for future work with pixel-based language models exploring appropriate strategies for learning morphological patterns (Klein and Tsarfaty, 2020; Seker and Tsarfaty, 2020; Soulos et al., 2021).

---

[16]Mean hidden state output across all tokens in a sentence, excluding the CLS token and black end-of-sequence token.

## 8 Conclusion

We evaluate four text rendering strategies to address the problem of redundancy in the input space of PIXEL-based language models. Consequently, more frequent parameter updates lead to better contextualised language representations. We find that rendering two characters per image patch (BIGRAMS) is a good trade-off between efficiency and generalisability, resulting in substantial improvements on downstream semantic and sentence-level tasks; contributing to open-vocabulary NLP with limited computational resources.

Further analyses reveal how the added rendering structure provokes clear representational similarities to what has been found in BERT. We see potential in future work generalising improvements found for tokenization-based masked language models to PIXEL-based masked language models. Furthermore, considering that the Vision Transformer has also been applied to speech modelling (Huang et al., 2022), and that patch representation has been suggested to be a critical component for the success of ViTs (Trockman and Kolter, 2023), we see potential for image patches as the basis for unifying modalities.

## Limitations

While the rendering strategies we propose here are well-suited to English, not all equally generalise to other languages or scripts. WORDS rendering relies on word boundaries which may not be readily available or well-defined for many languages which do not mark word or sentence boundaries with whitespace such as Thai or polysynthetic languages such as Inuktitut. MONO and BIGRAMS are more general approaches, but may affect the rendering of positional characters such as diacritics or correct contextual forms based on where boundaries are created. For both approaches, it may be necessary to modulate font size across languages to ensure character pairs fit into a single patch, especially when rendering with diacritics. MONO provides further representational efficiency compared to BIGRAMS by fixing character width, but comes at the cost of more limited language coverage; many scripts cannot be made fixed-width and fewer than 10 have mono fonts available. CONTINUOUS rendering provides a more general approach which must be balanced with learning efficiency.

## Acknowledgements

Jonas F. Lotz is funded by the ROCKWOOL Foundation (grant 1242). Elizabeth Salesky is supported by the Apple Scholars in AI/ML fellowship. Phillip Rust is funded by the Novo Nordisk Foundation (grant NNF 20SA0066568). This work was supported by a research grant (VIL53122) from VILLUM FONDEN.

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

# A  Appendix

## A.1  Representational efficiency

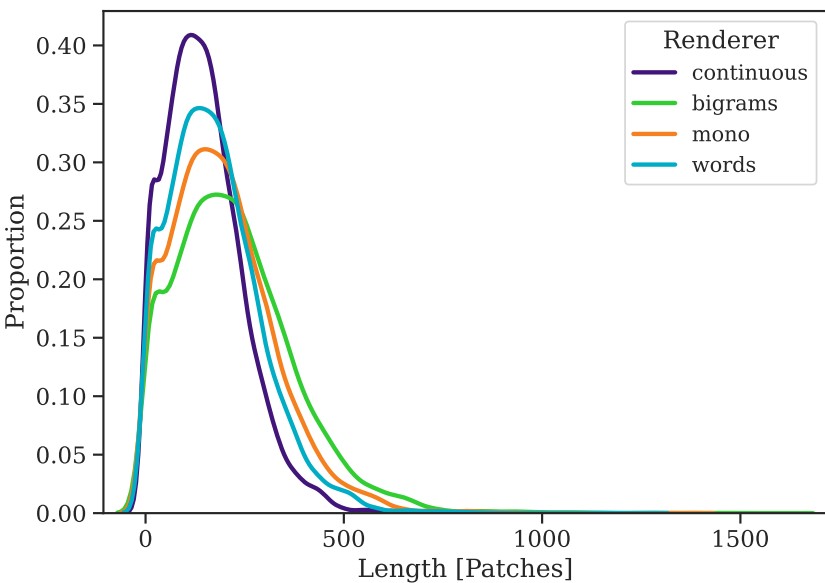

Figure 9: Distributions of sequence lengths (in patches) resulting from different rendering strategies.

As seen in Figure 1, structured rendering compresses the input space by reducing the positions characters may be observed in. This dramatically affects the number of unique inputs observed in a fixed number of sequences, as quantified in Figure 3. Concretely, the 10 most frequently observed image patches after processing 100,000 sequences from English Wikipedia are shown in Figure 2; with continuous rendering all are positional variants of the same subword, while with structured rendering each represents different words or morphemes. However, instituting word- or subword-level structure with whitespace padding increases sequence lengths compared to unstructured rendering as quantified in Figure 9.

## A.2  Pretraining loss curves

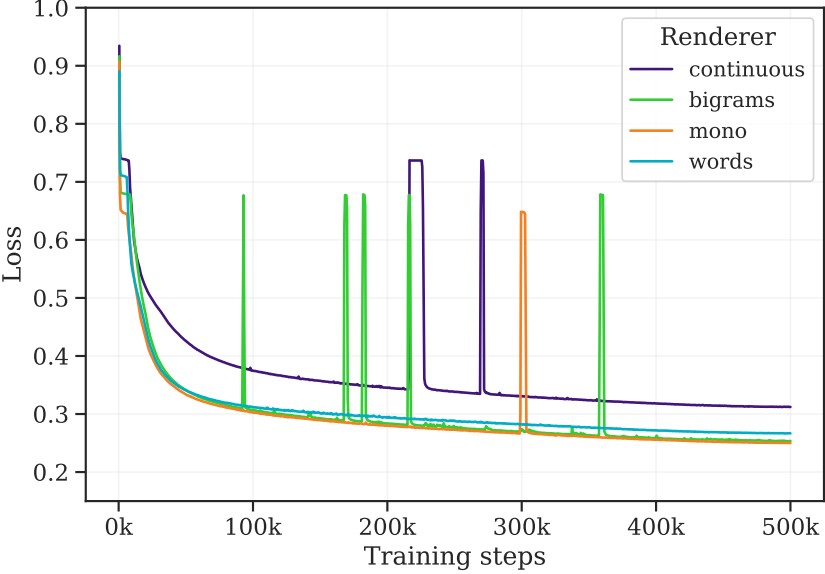

Figure 10: Pretraining loss for SMALL models with different rendering strategies, indicating that structured rendering may make the masked reconstruction task more data efficient, reaching a low loss in fewer steps.

## A.3   Detailed experimental results

| | ENG | ARA | COP | HIN | JPN | KOR | TAM | VIE | ZHO | AVG |
|---|---|---|---|---|---|---|---|---|---|---|
| BERT | 90.6 | 77.7 | 13.0 | 75.9 | 73.8 | 30.2 | 15.2 | 49.4 | 28.8 | 50.5 |
| PIXEL | 88.7 | 77.3 | 83.5 | 89.2 | 90.7 | 78.5 | 52.6 | 50.5 | 73.7 | 76.1 |
| TINY-CONTINUOUS | 78.9 | 74.6 | 80.0 | 87.9 | 89.9 | 75.1 | 48.3 | 46.2 | 69.5 | 72.3 |
| **Structure** | | | | | | | | | | |
| SMALL-CONTINUOUS | 87.2 | 77.2 | 83.4 | 88.9 | 91.0 | 78.8 | 53.8 | 51.9 | 73.5 | 76.2 |
| SMALL-BIGRAMS | 87.9 | 75.4 | 84.1 | 88.9 | 90.8 | 79.4 | 53.9 | 50.9 | 73.9 | 76.1 |
| SMALL-MONO | 88.3 | 76.8 | 83.4 | 88.9 | 91.0 | 79.0 | 50.5 | 51.3 | 73.8 | 75.9 |
| SMALL-WORDS | 88.0 | 77.2 | 83.9 | 89.3 | 91.2 | 78.7 | 53.7 | 53.3 | 74.2 | 76.6 |
| **Scale** | | | | | | | | | | |
| TINY-BIGRAMS | 82.9 | 70.6 | 79.1 | 86.2 | 90.0 | 76.2 | 44.9 | 47.6 | 69.8 | 72.0 |
| SMALL-BIGRAMS | 87.9 | 75.4 | 84.1 | 88.9 | 90.8 | 79.4 | 53.9 | 50.9 | 73.9 | 76.1 |
| BASE-BIGRAMS | 89.6 | 77.7 | 81.4 | 88.6 | 90.8 | 78.1 | 49.8 | 49.4 | 73.9 | 75.5 |

Table 4: Test set LAS results for dependency parsing on a selection of Universal Dependencies treebanks (UDP).

| | MNLI-M/MM | QQP | QNLI | SST-2 | COLA | STS-B | MRPC | RTE | WNLI | AVG |
|---|---|---|---|---|---|---|---|---|---|---|
| BERT | 84.0 / 84.2 | 87.6 | 91.0 | 92.6 | 60.3 | 88.8 | 90.2 | 69.5 | 51.8 | 80.0 |
| PIXEL | 78.1 / 78.9 | 84.5 | 87.8 | 89.6 | 38.4 | 81.1 | 88.2 | 60.5 | 53.8 | 74.1 |
| TINY-CONTINUOUS | 36.7 / 37.0 | 76.6 | 72.9 | 87.2 | 2.1 | 25.1 | 82.4 | 58.5 | 59.2 | 53.8 |
| **Structure** | | | | | | | | | | |
| SMALL-CONTINUOUS | 72.2 / 73.6 | 84.8 | 86.2 | 88.3 | 19.1 | 81.7 | 84.6 | 61.4 | 57.7 | 71.0 |
| SMALL-BIGRAMS | 77.3 / 78.1 | 85.7 | 87.8 | 90.4 | 42.3 | 84.3 | 87.8 | 63.5 | 56.3 | 75.4 |
| SMALL-MONO | 77.4 / 77.6 | 84.7 | 86.8 | 89.4 | 42.3 | 82.4 | 86.9 | 57.5 | 58.9 | 74.4 |
| SMALL-WORDS | 76.7 / 77.3 | 84.5 | 86.6 | 89.9 | 44.6 | 80.5 | 87.4 | 62.8 | 56.3 | 74.7 |
| **Scale** | | | | | | | | | | |
| TINY-BIGRAMS | 60.8 / 61.9 | 79.6 | 81.7 | 87.2 | 15.6 | 77.9 | 83.0 | 59.4 | 57.7 | 66.5 |
| SMALL-BIGRAMS | 77.3 / 78.1 | 85.7 | 87.8 | 90.4 | 42.3 | 84.3 | 87.8 | 63.5 | 56.3 | 75.4 |
| BASE-BIGRAMS | 81.1 / 81.4 | 87.6 | 89.7 | 90.4 | 53.3 | 86.6 | 90.2 | 63.5 | 56.3 | 78.0 |

Table 5: Validation set performance on GLUE. The reported metrics are $F_1$ score for QQP and MRPC, Matthew's correlation for COLA, Spearman's $\rho$ for STS-B, and accuracy for the rest.

|  | ENG | ARA | BEN | FIN | IND | KOR | RUS | SWA | TEL | AVG |
|---|---|---|---|---|---|---|---|---|---|---|
| BERT | 68.5 | 58.0 | 43.2 | 58.3 | 67.1 | 12.4 | 53.2 | 71.3 | 48.2 | 51.5 |
| PIXEL | 59.6 | 57.3 | 36.3 | 57.1 | 63.6 | 26.1 | 50.5 | 65.9 | 61.7 | 52.3 |
| TINY-CONTINUOUS | 42.6 | 45.0 | 12.4 | 45.3 | 48.1 | 13.2 | 36.7 | 46.8 | 45.7 | 36.6 |
| SMALL-CONTINUOUS | 57.1 | 53.3 | 20.3 | 57.5 | 62.9 | 22.3 | 51.1 | 65.3 | 58.1 | 48.8 |
| **Scale** | | | | | | | | | | |
| TINY-BIGRAMS | 43.3 | 45.5 | 19.0 | 50.3 | 48.2 | 14.9 | 45.4 | 52.7 | 56.4 | 41.6 |
| SMALL-BIGRAMS | 50.8 | 53.2 | 37.1 | 59.1 | 57.5 | 20.1 | 52.8 | 62.4 | 64.2 | 50.8 |
| BASE-BIGRAMS | 53.8 | 53.1 | 46.5 | 59.6 | 60.3 | 18.8 | 54.1 | 64.1 | 65.7 | 52.8 |

Table 6: Validation set $F_1$ scores for TyDiQA-GoldP. Average (AVG) scores exclude ENG (Clark et al., 2020). With some rendering structures, answer span extraction adversely affects results (see discussion at § A.4).

|  | AMH | HAU | IBO | KIN | LUG | LUO | PCM | SWA | WOL | YOR | AVG |
|---|---|---|---|---|---|---|---|---|---|---|---|
| BERT | 0 | 86.6 | 83.5 | 72.0 | 78.4 | 73.2 | 87.0 | 83.3 | 62.2 | 73.8 | 62.7 |
| PIXEL | 47.7 | 82.4 | 79.9 | 64.2 | 76.5 | 66.6 | 78.7 | 79.8 | 59.7 | 70.7 | 70.6 |
| BASE-BIGRAMS | 50.1 | 85.6 | 82.2 | 68.4 | 78.4 | 72.5 | 82.8 | 82.4 | 64.4 | 74.8 | 74.2 |

Table 7: Test set $F_1$ scores on MasakhaNER (Adelani et al., 2021). We follow the implementation of Rust et al. (2023) and render each word at the start of a new image patch.

## A.4 TyDiQa-GoldP

The CONTINUOUS rendering strategy used for PIXEL, in which words often overlap in an image patch, leads to extracted answer spans that potentially include leading or trailing characters that should not be part of the answer. BIGRAMS rendering adressess this issue by yielding clear word boundaries in the input representations.

However, the BIGRAMS rendering strategy poses new challenges to extracting answer spans for TyDiQA-GoldP. While the task is simplified compared to the primary task by removing language tracks that lack whitespace,[17] we find that a surprisingly high number of "words" are a string of comma-separated words or concatenations of characters and letters that should be delimited by whitespace. By design we consider and render these as one unit when we only split by whitespace. An example of a single "unit" from the training split highlights this issue more clearly: "oikeudet[1]Lääni[1]1**Vilna**523,0501387Vilnan"[18] where the expected answer is "**Vilna**" and highlighted in **bold**. In such an instance, a PIXEL BIGRAMS model will predict the whole unit, resulting in a lower performance. Furthermore, some of these "words" in the training data are more than a thousand characters long and therefore do not fit within the maximum sequence length of 529 patches.

---

[17]https://github.com/google-research-datasets/tydiqa/blob/master/gold_passage_baseline/README.md
[18]id = finnish-1438027099681899178-6

### A.5 Measuring self-similarity and intra-sentence similarity

We follow Ethayarajh (2019) and measure the degree of self-similarity and intra-sentence similarity for the words in the two frequency samples from § 6.3. Self-similarity is computed as the cosine similarity between the same word in different sentences and a high degree therefore indicates that representations vary little across contexts. For intra-sentence similarity we compute the cosine similarity between a word representation and the sentence representation (mean hidden state output across all tokens excluding the CLS token and black end-of-sequence token).[19] This captures how aligned the representation of a word is with the sentence as a whole. If a word has both a low degree of self-similarity and intra-sentence similarity, we infer that the word has a context-specific representation that is still distinct from the other words in that sentence. If self-similarity is low but intra-sentence similarity is high, this alludes to the word simply being contextualised by aligning its representation with the other words in that sentence. We summarise these two measures in Figure 7 and find that, just like in Figure 4a, the upper layers produce more context-specific representations as seen by the lower self-similarity, and that high-frequency words are the most context-specific. This is in line with Ethayarajh (2019) who finds that stopwords, being some of the most frequently observed words in the pretraining data, have some of the most context-specific representations. The measure of intra-sentence similarity reveals that the contextualised representation of low-frequency words is more similar to that of its context, with high-frequency words having more nuance where words do not necessarily mean the same just because they appear in the same sentence.

---

[19]Ethayarajh (2019) average over every word-sentence combination for a given sentence, not just a single word.