# OpenReview forum: "Text Rendering Strategies for Pixel Language Models"
_EMNLP/2023/Conference — EMNLP 2023 Main_

### Official Review · Reviewer_Tumj · 2023-08-03

**Soundness:** 3

**Excitement:**

3: Ambivalent: It has merits (e.g., it reports state-of-the-art results, the idea is nice), but there are key weaknesses (e.g., it describes incremental work), and it can significantly benefit from another round of revision. However, I won't object to accepting it if my co-reviewers champion it.

**Paper Topic And Main Contributions:**

The author proposes that for the current pixel-based language model, suboptimal almost-equivalent input patches may be generated when processing downstream tasks. The author further found through research that a simple character bigram structure can effectively improve the performance of the sentence-level task. The new rendering strategy proposed by the author can also make the 22M language model achieve performance equivalent to the 86M language model.

**Questions For The Authors:**

1.	The authors mentioned that they performed experiments on UD 2.10 and the experimental results are given in the Appendix. However, the authors’ method is to add a simple task-specific MLP to the language model. This is not suitable for parsing tasks on UD, because directly using classification MLP for parsing will lead to illegal structures (such as unrooted trees or ring structures), and additional post-processing is required in parsing Steps to remove this illicit influence. Generally speaking, post-processing will reduce the performance of the model. Is the author's results in Table 4 in the appendix post-processing? If not, then this performance is not of reference significance.
2.	To achieve parsing by classification, two different classifier heads are generally required because two different label results of head and relation are to be generated. Neither the traditional transition-based method nor the graph-based method can use a classification head to achieve parsing. I am very curious how the author achieved it (The simplest structure of parsing using the masked language model still requires a specially designed Biaffine structure header and chu-liu post-processing algorithm, see [1] for details)? In addition, corresponding to the accuracy of head and relation, two statistical indicators, UAS and LAS, are generally used in parsing, but the results in Table 4 do not indicate which indicator is used at all. In addition, the parsing task generally uses 12 fixed languages in UD 2.0 or UD 2.2 (Bulgarian, Catalan, Czech, German, English, Spanish, French, Italian, Dutch, Norwegian, Romanian, Russian[2]). There are many baselines for reference in these languages, but the author chose UD 2.10, and did not use the common 12 languages, so it is impossible to make an accurate judgment on the performance comparison in Table 4.
[1] Establishing Strong Baselines for the New Decade: Sequence Tagging, Syntactic and Semantic Parsing with BERT. Han He and Jinho D.Choi. AAAI 2020.
[2] Stack-pointer Networks for Dependency Parsing. Xuezhe Ma, Xecong Hu, Jingzhou Liu, Nanyun Peng, Graham Neubig, and Eduard Hovy. ACL 2018.

**Reasons To Accept:**

1.	The rendering strategy proposed by the author can make the small model achieve the same effect as the large model, and can be used as a means of compressing model parameters
2.	The authors’ ablation experiment design is very detailed, and the experimental results are analyzed in terms of model parameter magnitude, structure comparison, token frequency, etc., which is very detailed.

**Reasons To Reject:**

1.	The novelty of this paper is slightly insufficient. The new rendering strategy proposed by the author is an improvement of the rendering strategy of the previous pixel-based language model.
2.	The experimental results in this paper, especially the relevant parts of UDP, may have some problems (see the Questions section for details)

**Reproducibility:**

3: Could reproduce the results with some difficulty. The settings of parameters are underspecified or subjectively determined; the training/evaluation data are not widely available.

**Reviewer Confidence:**

4: Quite sure. I tried to check the important points carefully. It's unlikely, though conceivable, that I missed something that should affect my ratings.

**Typos Grammar Style And Presentation Improvements:**

The last line in Table 2 is displayed in light gray instead of normal black. I didn't find the corresponding explanation or emphasis from authors. This seems to be a formatting error.

---

> ### Author Rebuttal · Authors · 2023-08-28
>
> We thank the reviewer for your time and feedback. We are pleased to see that you appreciate the detailed analyses, and that you agree with the importance of building compact and resource-friendly models. To address your concerns:
>
> > The novelty of this paper is slightly insufficient. The new rendering strategy proposed by the author is an improvement of the rendering strategy of the previous pixel-based language model.
>
> While character (n=2)-gram models are not novel, as discussed in the Related work section (L535-546), hypothesis-driven improvements through new text rendering strategies are. All previous work on pixel-based language modelling [1,2,3,4] used the continuous rendering strategy without considering any alternative approaches. To the best of our knowledge this is also the first work to investigate the embedding space of pixel-based language models and identify representational similarities to tokenization-based language models. We regard this new insight as valuable also for future work in this direction because it opens up new avenues for further improving pixel-based models, given the accumulated knowledge from techniques that have proven successful from tokenization-based models (L506-510; L570-575).
>
> > The experimental results in this paper, especially the relevant parts of UDP, may have some problems (see the Questions section for details)
>
> We acknowledge the reviewer’s concern. To be clear, we follow the same protocol during finetuning as done in PIXEL [3]. This is stated on line 248-249. From [3, page 4]:
> “*For dependency parsing, we render text as above but obtain word-level representations by mean pooling over all corresponding image patches of a word and employ a biaffine parsing head (Dozat & Manning, 2017), following the implementation from Glavaš & Vulic (2021).*”
>
> > However, the authors’ method is to add a simple task-specific MLP to the language model.
>
> We understand this concern, which is directly related to the question above. We will rework section 5.2 to make our approach clear and avoid confusion about how the model is finetuned for the dependency parsing task. We also acknowledge that we have a typo and that line 241-243 should read: “*To finetune our models for classification tasks we replace the decoder used for pretraining with a task-specific classification head.*”, dropping the erroneous mention of MLP.
>
> > To achieve parsing by classification, two different classifier heads are generally required because two different label results of head and relation are to be generated. Neither the traditional transition-based method nor the graph-based method can use a classification head to achieve parsing. I am very curious how the author achieved it
>
> See response above. We use a biaffine parsing head from Dozat & Manning (2017), following the implementation from Glavaš & Vulic (2021).
>
> > In addition, corresponding to the accuracy of head and relation, two statistical indicators, UAS and LAS, are generally used in parsing, but the results in Table 4 do not indicate which indicator is used at all.
>
> We report test set LAS scores. This will be clearly stated in the final version of the paper.
>
> > In addition, the parsing task generally uses 12 fixed languages in UD 2.0 or UD 2.2 (Bulgarian, Catalan, Czech, German, English, Spanish, French, Italian, Dutch, Norwegian, Romanian, Russian[2]).
>
> We agree that the choice of languages for multilingual evaluation is an important issue.
> In this paper, we re-used the same set of languages as [3] for the sake of direct comparison. There is no straightforward agreement on which languages should be used for multilingual evaluations of language models [5], so re-using the languages from [3] means that our experiments cover 8 different writing scripts: Latin, Chinese, Japanese, Korean, Arabic, Devanagari, Coptic, and Tamil; as opposed to only covering Latin and Cyrillic in the 12 fixed languages. Nevertheless, if the reviewer still thinks this would strengthen the contribution of the paper, we can include the results on the 12 fixed languages in the appendix of final version of the paper. We are currently not able to present these results due to limited computing resources during the author response period.
>
> > The last line in Table 2 is displayed in light gray instead of normal black. I didn't find the corresponding explanation or emphasis from authors. This seems to be a formatting error.
>
> We thank the reviewer for highlighting this problem, which we will address when we revise the paper.
>
> > Reproducibility.
>
> We aim for our work to be fully reproducible. We emphasise that we will make all code and models available to the public (L87-88). All details about (hyper)parameter values can be found in [3], since we aimed for a clean comparison by varying as few aspects as possible. The exception being the learning rates, which are stated in footnote 5 on page 4.
>
> [1]	Elizabeth Salesky, David Etter, and Matt Post. 2021. Robust open-vocabulary translation from visual text representations. *EMNLP*.
>
> [2]	Elizabeth Salesky, Neha Verma, Philipp Koehn, and Matt Post. 2023. Pixel representations for multilingual translation and data-efficient cross-lingual transfer. *arXiv preprint*.
>
> [3]	Phillip Rust, Jonas F. Lotz, Emanuele Bugliarello, Elizabeth Salesky, Miryam de Lhoneux, and Desmond Elliott. 2023. Language modelling with pixels. *ICLR*.
>
> [4]	Michael Tschannen, Basil Mustafa, and Neil Houlsby. 2023. Image-and-language understanding from pixels only. *CVPR*.
>
> [5]	Ethan A. Chi, John Hewitt, and Christopher D. Manning. 2020. Finding Universal Grammatical Relations in Multilingual BERT. *ACL*.

---

### Official Review · Reviewer_u9Mz · 2023-08-04

**Soundness:** 3

**Excitement:**

2: Mediocre: This paper makes marginal contributions (vs non-contemporaneous work), so I would rather not see it in the conference.

**Paper Topic And Main Contributions:**

This paper evaluates four text rendering strategies to address the problem of redundancy in the input space of PIXEL-based language models.

**Reasons To Accept:**

This paper evaluates four text rendering strategies to address the problem of redundancy in the input space of PIXEL-based language models. Pixel-based language models process text rendered as images, making them a promising approach to open vocabulary language modelling.

**Reasons To Reject:**

1. The methods proposed in this paper are relatively simple and lack novelty.
2. While the rendering strategies this paper proposed are suit to English, those strategies may not generalize to other languages.


**Reproducibility:**

3: Could reproduce the results with some difficulty. The settings of parameters are underspecified or subjectively determined; the training/evaluation data are not widely available.

**Reviewer Confidence:**

2: Willing to defend my evaluation, but it is fairly likely that I missed some details, didn't understand some central points, or can't be sure about the novelty of the work.

---

> ### Author Rebuttal · Authors · 2023-08-28
>
> We thank the reviewer for your time and feedback. To address your concerns:
>
> > The methods proposed in this paper are relatively simple and lack novelty.
>
> While character (n=2)-gram models are not novel, as discussed in the Related work section (L535-546), hypothesis-driven improvements through new text rendering strategies are. All previous work on pixel-based language modelling [1,2,3,4] used the continuous rendering strategy without considering any alternative approaches. We see it as a strength that the downstream task improvements can be realised from a straightforward modification of the rendering strategy that generalises across scripts and languages.
>
> > While the rendering strategies this paper proposed are suit to English, those strategies may not generalize to other languages.
>
> We agree that the rendering strategies may appear to be suited to English but they were not designed with latin script writing systems in mind. The paper includes both a discussion of this in the Limitations section (L574-596) and experiments on TyDiQa-GoldP (L305-310) to specifically address the concern that the bigrams rendering strategy may not generalise to other languages. For languages that do not mark word or sentence boundaries with whitespace such as Thai or polysynthetic languages such as Inuktitut, the bigrams text rendering strategy would still result in a compressed input space (increase in observed token frequency) compared to a continuous rendering strategy.
> The TyDiQa-GoldP benchmark already includes typologically diverse languages: Arabic, Bengali, Finnish, Indonesian, Kiswahili, Korean, Russian, and Telugu (4/8 being Latin scripts). We find that the bigrams rendering strategy leads to better performance compared to the PIXEL baseline (52.8 vs 52.3) and with the smaller model scales benefitting even more (50.8 vs 48.8 for SMALL; 41.6 vs 36.6 for TINY).
>
> Additionally, in response to your concern, we have also finetuned the BASE-BIGRAMS model for NER on MasakhaNER [5] during this response period. We find that the proposed bigrams rendering strategy leads to consistent improvements across the 10 non-English languages (Amharic, Hausa, Igbo, Kinyarwanda, Luganda, Luo, Nigerian-Pidgin, Swahili, Wolof, Yorùbá) compared to PIXEL following a continuous rendering strategy (74.2 vs 70.6). We will include the results of this experiment when we prepare the final version of the paper.
> |               | Amharic  | Hausa  | Igbo  | Kinyarwanda  | Luganda  | Luo  | Nigerian-Pidgin  | Swahili  | Wolof  | Yorùbá  | AVG  |
> |---------------|------|------|------|------|------|------|------|------|------|------|------|
> | BASE-BIGRAMS   | 50.1 | 85.6 | 82.2 | 68.4 | 78.4 | 72.5 | 82.8 | 82.4 | 64.4 | 74.8 | 74.2 |
> | PIXEL          | 47.7 | 82.4 | 79.9 | 64.2 | 76.5 | 66.6 | 78.7 | 79.8 | 59.7 | 70.7 | 70.6 |
>
>
> > Reproducibility.
>
> We aim for our work to be fully reproducible. We emphasise that we will make all code and models available to the public (L87-88). All details about (hyper)parameter values can be found in [3], since we aimed for a clean comparison by varying as few aspects as possible. The exception being the learning rates, which are stated in footnote 5 on page 4.
>
> [1]	Elizabeth Salesky, David Etter, and Matt Post. 2021. Robust open-vocabulary translation from visual text representations. *EMNLP*.
>
> [2]	Elizabeth Salesky, Neha Verma, Philipp Koehn, and Matt Post. 2023. Pixel representations for multilingual translation and data-efficient cross-lingual transfer. *arXiv preprint*.
>
> [3]	Phillip Rust, Jonas F. Lotz, Emanuele Bugliarello, Elizabeth Salesky, Miryam de Lhoneux, and Desmond Elliott. 2023. Language modelling with pixels. *ICLR*.
>
> [4]	Michael Tschannen, Basil Mustafa, and Neil Houlsby. 2023. Image-and-language understanding from pixels only. *CVPR*.
>
> [5]	David Ifeoluwa Adelani, Jade Abbott, Graham Neubig, Daniel D’souza, Julia Kreutzer, Constantine Lignos, Chester Palen-Michel, Happy Buzaaba, Shruti Rijhwani, Sebastian Ruder, Stephen Mayhew, Israel Abebe Azime, Shamsuddeen H. Muhammad, Chris Chinenye Emezue, Joyce Nakatumba-Nabende, Perez Ogayo, Aremu Anuoluwapo, Catherine Gitau, Derguene Mbaye, Jesujoba Alabi, Seid Muhie Yimam, Tajuddeen Rabiu Gwadabe, Ignatius Ezeani, Rubungo Andre Niyongabo, Jonathan Mukiibi, Verrah Otiende, Iroro Orife, Davis David, Samba Ngom, Tosin Adewumi, Paul Rayson, Mofetoluwa Adeyemi, Gerald Muriuki, Emmanuel Anebi, Chiamaka Chukwuneke, Nkiruka Odu, Eric Peter Wairagala, Samuel Oyerinde, Clemencia Siro, Tobius Saul Bateesa, Temilola Oloyede, Yvonne Wambui, Victor Akinode, Deborah Nabagereka, Maurice Katusiime, Ayodele Awokoya, Mouhamadane MBOUP, Dibora Gebreyohannes, Henok Tilaye, Kelechi Nwaike, Degaga Wolde, Abdoulaye Faye, Blessing Sibanda, Orevaoghene Ahia, Bonaventure F. P. Dossou, Kelechi Ogueji, Thierno Ibrahima DIOP, Abdoulaye Diallo, Adewale Akinfaderin, Tendai Marengereke, and Salomey Osei. MasakhaNER: Named Entity Recognition for African Languages. *TACL*.

---

### Official Review · Reviewer_ocCw · 2023-08-07

**Soundness:** 5

**Excitement:**

4: Strong: This paper deepens the understanding of some phenomenon or lowers the barriers to an existing research direction.

**Paper Topic And Main Contributions:**

This papers focuses on the comparison of four rendering strategies for pixel-based language models. It contributes an empirical comparison of said rendering strategies on three different datasets (UDP, GLUE, and TyDiQA-GoldP), finding character bigram-based structure to be considerably more computationally efficient than previous work (the continuous structure). The authors further provide an ablation study and an investigation of the embedding space that reveals an anisotropy (low-frequency tokens being clustered together).

**Reasons To Accept:**

The paper analyses its points thoroughly. The bigram representation proposed in the paper achieves competitive results with fewer parameters than previous pixel-based representations. It is important to encourage such efforts for more efficient NLP given the environmental impact of resource-hungry models.

**Reasons To Reject:**

The advantages of pixel-based language models over tokenization-based ones are only explicitly highlighted in the introduction, and the proposed pixel-based models do not perform better than tokenization-based ones yet in settings with sufficient language resources.

**Reproducibility:**

4: Could mostly reproduce the results, but there may be some variation because of sample variance or minor variations in their interpretation of the protocol or method.

**Reviewer Confidence:**

4: Quite sure. I tried to check the important points carefully. It's unlikely, though conceivable, that I missed something that should affect my ratings.

**Typos Grammar Style And Presentation Improvements:**

My suggestion would be to highlight the performance of pixel-based models (at least for BIGRAMS) on low-resource languages in the main text, not just in the appendix.

---

> ### Author Rebuttal · Authors · 2023-08-28
>
> We thank the reviewer for your time and positive feedback on our work. We are grateful to see that you appreciate the thoroughness of our analyses, and that you agree with the importance of building compact and resource-friendly models. We will definitely highlight the performance of the pixel-based models on low-resource languages in the main text. We hope to see the field of pixel-based language modelling mature just as tokenization-based models have.

---

### Meta-Review · Area_Chair_zQ8V · 2023-09-29

**Recommendation:** 4

**Metareview:**

This paper presents a comparative analysis of various rendering algorithms utilized in pixel-based language models. The reviewers acknowledged the level of detail in the study and commended the efficiency of the methods employed. They also emphasized the need for further research in this area, considering the growing interest in multimodal language models.

---

### Decision · Program_Chairs · 2023-10-07

**Decision:**

Accept-Main

**Comment:**

This paper presents a comparative analysis of various rendering algorithms utilized in pixel-based language models. The reviewers acknowledged the level of detail in the study and commended the efficiency of the methods employed. They also emphasized the need for further research in this area, considering the growing interest in multimodal language models.